# Real-World, Multicenter Case Series of Patients Treated with Isavuconazole for Invasive Fungal Disease in China

**DOI:** 10.3390/microorganisms11092229

**Published:** 2023-09-04

**Authors:** Lisha Wu, Shougang Li, Weixi Gao, Xiaojian Zhu, Pan Luo, Dong Xu, Dong Liu, Yan He

**Affiliations:** 1Department of Pharmacy, Tongji Hospital, Tongji Medical College, Huazhong University of Science and Technology, Wuhan 430030, China; wulisa_happy@163.com (L.W.); luopan106@126.com (P.L.); 2Department of Pharmacy, Huangjiahu Hospital, Hubei University of Traditional Chinese Medicine, Wuhan 430065, China; 3Department of Rehabilitation Medicine, General Hospital of Central Theater Command of Chinese People’s Liberation Army, Wuhan 430070, China; shgangli@outlook.com; 4Department of Pharmacy, Hubei General Hospital, Renmin Hospital of Wuhan University, Wuhan 430060, China; gaoweixi@whu.edu.cn; 5Department of Hematology, Tongji Hospital, Tongji Medical College, Huazhong University of Science and Technology, Wuhan 430030, China; zhuxiaojian@hust.edu.cn; 6Department of Infection Disease, Tongji Hospital, Tongji Medical College, Huazhong University of Science and Technology, Wuhan 430030, China; xdong@tjh.tjmu.edu.cn

**Keywords:** isavuconazole, invasive fungal disease, clinical response, efficacy, safety

## Abstract

Background: The incidence of invasive fungal disease (IFD) has increased significantly, and IFD is a major cause of mortality among those with hematological malignancies. As a novel second-generation triazole antifungal drug offering both efficacy and safety, isavuconazole (ISA) is recommended by various guidelines internationally for the first-line treatment of invasive aspergillosis (IA) and invasive mucormycosis (IM) infecting adults. Given that it was only approved in China at the end of 2021, there is currently a lack of statistical data regarding its usage in the Chinese population. The primary objective of this report is to describe early experiences with ISA for the treatment of IFD. Methods: This was a real-world, multicenter, observational case series study conducted in China. It included patients from three centers who received ISA treatment from January 2022 to April 2023. A retrospective assessment on patient characteristics, variables related to ISA administration, the treatment response of IFD to ISA, and potential adverse events attributed to ISA was conducted. Results: A total of 40 patients met the inclusion criteria. Among them, 12 (30%) were diagnosed with aspergillosis, 2 (5%) were diagnosed with candidiasis, 12 (30%) were diagnosed with mucormycosis, and 14 cases did not present mycological evidence. The predominant site of infection was the lungs (36), followed by the blood stream (8), sinuses (4), and respiratory tract (2). The overall response rate was 75% (30 patients), with male patients having a higher clinical response than female patients (24/24 versus 6/16, *p* = 0.000) and autologous stem cell transplant patients having a higher clinical response than allogeneic stem cell transplant patients (6/6 versus 4/10, *p* = 0.027). During the observation period, four patients experienced adverse effects associated with ISA, but none of them discontinued the treatment. Conclusions: Our findings suggest that ISA, a novel first-line treatment for IA and IM, is associated with a high clinical response rate, low incidence, and a low grade of adverse effects. Given the short time that ISA has been available in China, further research is needed to identify its efficacy and safety in the real world.

## 1. Introduction

IFD (invasive fungal disease), which generally occurs in immunocompromised populations (individuals with hematologic malignancies, individuals who have had solid organ or stem cell transplants, HIV-infected individuals, etc.), refers to diseases in which fungi invade the human body; grow and multiply in tissues, organs, or blood; and cause inflammatory reactions and tissue damage. IFDs are a major cause of morbidity and mortality among immunocompromised patients with hematologic malignancies (HMs) and who have undergone stem cell transplants [1]. They pose a critical threat to public health and are an under-recognized component of antimicrobial resistance, an emerging crisis worldwide [2]. The most common pathogenic fungi are *Candida* and *Aspergillus*, while *Mucor* is rare. Fungalemia is more commonly caused by *Candida*. Pulmonary IFD is predominantly caused by filamentous fungi, of which *Aspergillus* is the main pathogenic bacteria [3]. It is worth noting that the infection rate of *Mucor* has gradually increased in recent years due to the use of triazole prophylaxis. The poor prognosis of IFD patients is attributable to difficulties in diagnosing IFD as well as the emergence of strains resistant to traditional triazoles and the elevated incidence of adverse effects brought on by amphotericin B (AmB) [3,4], which have caused great trouble for clinicians. Therefore, there is an urgent need to introduce a drug with a broad antimicrobial spectrum, high safety properties, excellent tolerability, and few drug interactions.

As a novel second-generation triazole, ISA was approved by the US FDA in 2015 as the first-line agent for invasive aspergillosis (IA) and invasive mucormycosis (IM) infecting adults, and it has also been recommended by certain guidelines internationally for the treatment of IA and IM [5,6,7,8,9,10,11]. ISA inhibits the cytochrome lanosterol 14α-demethylase (CYP51) and destroys the structure and function of the fungal cell membrane by blocking the synthesis of ergosterol on the fungal cell membrane [12]. A special side chain of the ISA molecule has a strong affinity for the CYP51 protein, which endows it with a broad antifungal spectrum and efficacy against fungi resistant to other triazoles. ISA is currently available in capsule and intravenous formulations. Unlike the injectable forms of voriconazole (VOR) and posaconazole (POS), the water-soluble, intravenous form of ISA sulfate is not supplemented with the excipient sulfobutylether-β-cyclodextrin (SBECD) [12]. Furthermore, its oral absorption is not affected by food and stomach acid. As a consequence, ISA has provided the possibility to shift from an intravenous to an oral route and a once-daily frequency of administration, thus facilitating treatment compliance [13] as well as broadening the patient population to which it can be administered. Due to the results of a case–control trial [14], ISA became the sole first-line treatment approved by the FDA for IM therapy aside from AmB [9].

At present, the efficacy and safety of ISA have mainly been reported in clinical trials, rarely in real-world studies. Most of these studies have shown that ISA is not inferior to VOR and AmB with respect to the treatment of IFD, presenting a 35–87% clinical response and a 6–38% 42-day mortality [15,16,17,18,19,20,21]. Given that it was only approved in China at the end of 2021 for the treatment of IA and IM in adults, there is a lack of real-world evidence concerning the Chinese population. Therefore, the primary objective of this report was to describe the early Chinese experience with ISA for the treatment of IFD.

## 2. Materials and Methods

### 2.1. Patient Population

This was a real-world, multicenter, observational case series study conducted at three medical centers in Hubei, China, between January 2022 and April 2023. Patients receiving ISA for the treatment of IFD were included. The included cases were treated with ISA in inpatient and/or outpatient settings. Cases where treatment status could not be assessed due to incomplete medical records were excluded from the study. The definition of clinical success was a favorable response to the treatment, which was assessed on Day 42 after initiating the ISA therapy. If the ISA treatment was not administered for a full 42 days, the assessment would be conducted at the end of the ISA treatment. Clinical improvement included a complete response and a partial response (improvement in clinical/radiologic findings but not complete resolution). We also aimed to describe the incidence of adverse effects as recorded in the patients’ electronic medical records.

The study protocol was approved by the institutional review board of each participating center prior to data collection. The study did not include factors that would have required patient consent.

### 2.2. Definitions

The revised 2020 European Organization for Research and Treatment of Cancer and the Mycoses Study Group Education and Research Consortium (EORTC/MSGERC) criteria [22] were adopted as the diagnostic criteria for IFD. A proven IFD was defined as the presence of documented histopathologic and microbiological evidence of IFD in a tissue biopsy or needle aspiration specimen obtained from a normally sterile site, excluding samples taken from BAL, cranial sinus cavity, and urine. A probable IFD was defined as the presence of at least one microbiological criterion (cytology, culture, or positive GM test results) along with one host factor (recent absolute neutrophil count, ANC < 500 cells/mL, allogeneic stem cell transplant, T-cell immune suppressant therapy, or prolonged corticosteroid use) and one clinical criterion (nodule, cavitary, or ground-glass opacities found via computed tomography (CT); tracheobronchitis; or sinonasal infection). A possible IFD was defined as the requirement for a host factor along with a radiological criterion (nodule, cavitary, and ground-glass opacities observed via CT).

According to the criteria for the efficacy of IFD treatment [1,23], breakthrough infection was defined as the development of a new definite, probable, or possible IFD after at least 7 days of prophylaxis with an antifungal treatment with adequate anti-mold coverage. A favorable response included the complete or partial resolution of clinical, radiologic, and microbiological findings. A complete response was defined as a complete clearance of clinical signs and symptoms, radiologic lesions identified earlier via chest X-ray or CT, and all relevant microbiological findings. Partial response was defined as a clinically significant improvement in signs and symptoms, improvement in radiologic abnormalities, and no related microbiological findings. Targeted treatment was defined as the use of ISA to treat the patients with proven/probable IFD. Empirical treatment was defined as the use of ISA to treat the patients with possible IFD. Safety outcomes included adverse events during treatment and the percentage of premature discontinuation of ISA. Possible adverse events associated with ISA included changes in liver function, kidney function, and electrocardiogram findings before and after the application of ISA. The level of adverse effects was assessed according to Common Terminology Criteria for Adverse Events (Version 5.0). The death of a patient and the abandonment of treatment by the patient were classified as instances of treatment failure.

### 2.3. Statistical Analysis

Fisher’s test and Mann–Whitney U test were used for tests of differences for categorical and continuous variables, respectively. Significance tests were two-sided, and *p*-value ≤ 0.05 was considered statistically significant. Data analysis was performed using SPSS 26.0.

## 3. Results

### 3.1. Characteristics of Patients Included in the Study

A total of 40 patients were treated with ISA between January 2022 and April 2023, and the demographics of these patients are summarized in Table 1. A flowchart regarding the patients included and excluded in the analysis is shown in Figure 1. The characteristics of patients with IFD are listed in Table 2.

The median age and weight of patients were 46 years (9–72) and 59 kg (35–96), respectively, and 16 patients (40%) were female, most of whom presented hematologic malignancies (28/40, 70.0%). The most common site of infection was the lungs (36/40), followed by the bloodstream (8/40), sinuses (4/40), and respiratory tract (2/40). There were 16 patients with a history of stem cell transplantation (6 autologous and 10 allogeneic); other common comorbidities included diabetes mellitus (10/40), hepatic insufficiency (8/40), renal insufficiency (16/40), graft-versus-host disease (6/40), and respiratory failure (4/40).

### 3.2. Treatment of ISA and Occurrence of IFD

Overall, there were 30 (75%) patients who achieved a clinical response, and clinical responses were higher among male patients than female patients (24/24 vs. 6/16, *p* = 0.000, Table 3). Clinical responses were higher for patients who had had an autologous stem cell transplant than they were for allogeneic stem cell transplant patients (6/6 vs. 4/10, *p* = 0.027, Table 3). The response rates were 66.7% (8/12), 0% (0/2), and 83.3% (10/12) for patients with IA, IC, and IM, respectively. Treatments provoked a response for 100% (2/2), 66.7% (16/24), and 85.7% (12/14) of patients with proven, probable, and possible fungal infections, respectively. In terms of the site of infection, the clinical response rates of the patients with pulmonary infection, sinus infection, respiratory tract infection, and bloodstream infection were 72.2% (26/36), 100% (4/4), 100% (2/2), and 75% (6/8), respectively. After treatment, 26 patients who had positive cultures before treatment yielded negative culture results. See Figure 2 for details.

IA affected 12 patients with hematological malignancies, invasive candidiasis (IC) occurred in 2 patients with hematological malignancies, and IM infected 12 patients (5 with hematological malignancies). The remaining 14 (35%) patients failed to receive an etiological diagnosis.

### 3.3. Side Effects

Four (10%) patients experienced adverse effects associated with orally administered ISA, all of whom had grade 2 gastrointestinal disorders. However, they did not abandon the ISA treatment, because the adverse effects disappeared after switching to intravenous administration and rehydration. In addition, the ISA treatments were clinically effective for all patients.

## 4. Discussion

We conducted this study due to the fact that ISA was launched in China at the end of 2021 and to address the lack of real-world data for the Chinese population. This study also included two pediatric patients with hematological malignancies, thus filling the gap in the data for patients in this age group.

Previous reports showed that ISA had become a new option for the treatment of IFD, with an overall response rate of 35–87% [15,16,17,18,19,20,21]. In our study, we had an overall response rate of 75%. In terms of infection sites, most patients had lung infections (90%, 36/40), for which the response rate was 72.2% (26/36), which is similar to the result reported by Cattaneo et al. [17], and sinus infection, respiratory tract infection, and bloodstream infection had response rates of 100.0% (4/4), 100.0% (2/2), and 75.0% (6/8), respectively. Regarding pathogens, the clinical response rates were different, with 66.7% (8/12) of patients infected with IA, 0% (0/2) of patients infected with IC, and 83.3% (10/12) of patients infected with IM. In our study, the main types of infections were IA and IM, and the incidence of IM tended to increase, which is consistent with previous reports [3,24].

A series of studies [25,26,27] demonstrated that there were no significant differences between VOR or POS prophylaxis and ISA prophylaxis with respect to the incidence or mortality of breakthrough IFD among patients with hematological oncology and allogeneic hematopoietic stem cell transplantation. Data from another study [28] suggest that ISA is sufficiently efficient and secure for the treatment of patients after the failure of other azole and non-azole prophylaxis. Studies have shown that the emergence of resistant strains and inadequate drug concentrations are common probable causes of breakthroughs [29]. The non-attainment of plasma concentrations of POS due to various factors is considered a major risk factor for breakthrough IM [30], especially with POS suspension [7]. In our study, Case 2 was a patient with hemophagocytosis who did not experience breakthrough IFD during ISA prophylaxis. However, two weeks after discontinuing ISA treatment and switching to POS prophylaxis, the patient developed breakthrough IM and, ultimately, discontinued treatment due to septic shock.

ISA is not approved for the treatment of pediatric patients, but clinical studies have been published regarding the treatment of children with IA and IM. A foreign case series showed that ISA may be useful and safe for children with hematological malignancies, even in the case of stem cell transplantation [13]. Based on a meta study [31], among children who do not tolerate or respond to pre-existing indications, it is necessary to weigh the pros and cons of ISA administration before deciding whether to use it for treatment. According to the literature, pediatric patients should be administered a loading dose of 10 mg/kg q 8 h for 2 days and a maintenance dose of 10 mg/kg qd, with a maximum dose not exceeding that of adults [9]. In our study, two pediatric patients (9 and 16 years old) with acute myeloid leukemia who received off-label empirical treatment with ISA in adult doses achieved good results; these results enrich the data on the clinical experience of administering ISA to Chinese children. Although therapeutic drug monitoring of ISA among adults is currently considered controversial, the monitoring of blood concentrations and adverse effects is necessary due to the more rapid clearance of ISA in children [9,13,32].

Our study indicates that patients receiving ISA treatment, especially elderly patients, did not experience significant adverse reactions or drug interactions. Additionally, five patients switched to ISA due to experiencing adverse effects of VOR or AmB, and their clinical symptoms were relieved. Four patients experienced grade 2 adverse reactions (gastrointestinal), which were resolved by changing dosage forms and through clinical fluid replacement. The results of the SECURE [15] and VITAL [21] trials showed that ISA was not inferior to VOR and AmB with respect to the treatment of invasive mycosis, presenting better tolerability and fewer drug-related adverse events, thus favoring the use of ISA as the primary agent for treating IFD. Hamed et al. [33] conducted a post-hoc analysis of the SECURE and VITAL trials, and the results showed that patients over 65 had worse clinical outcomes than those under 65. Older patients are usually on multiple medications, so the concomitant application of triazoles might cause serious drug–drug interactions (DDIs). As with other triazoles, ISA is a substrate for cytochrome P450 (CYP3A4); CYP3A4 inducers such as rifampicin, carbamazepine, and long-acting barbiturates reduce the concentration of ISA. Remi Pieragostini et al. [34] conducted a single-center study in France showing that the blood concentration of ISA was generally higher than the recommended therapeutic reference range regardless of the patient’s basic condition and did not cause serious adverse effects. Meanwhile, despite the fact that ISA is a moderate inhibitor of CYP3A4, Groll et al. [35] believe that ISA has less of an effect on the plasma concentrations of vulnerable drugs such as cyclosporine compared to VOR and POS. Thus, with respect to reducing the occurrence of DDIs, ISA is a valuable addition to other triazole antifungal regimens [36,37], especially for older patients with IFD.

In our study, four patients experienced grade 2 adverse reactions (gastrointestinal), which were resolved via changing dosage forms and clinical fluid replacement. Generally speaking, the most common adverse effects associated with ISA are hepatic abnormalities and gastrointestinal disorders, including nausea, vomiting, and diarrhea, which are mild symptoms and rarely lead to the drug’s discontinuation [16,18]. However, it is still worth noting that ISA might shorten QT interval duration, unlike other triazoles, which might cause QT interval prolongation [38], and should be avoided for patients with a family history of short QT intervals. Dipippo et al. [39] retrospectively analyzed 23 patients with leukemia, of which 20 patients developed POS-related hepatotoxicity, and after switching to ISA, elevated aminotransferases rapidly decreased to normal levels and hepatic function recovered. Three patients died during the observation period. Patient 1 died from tumor recurrence and central nervous system involvement leading to central respiratory failure. Patient 6 died from respiratory failure caused by severe intracranial infection after an allogeneic transplantation. Despite standardized treatment, Patient 13 had difficulty maintaining respiration and circulation, and his death was inevitable.

The Chinese Assessment of Antifungal Therapy in Hematological Disease study showed [40] that after 6 months, the incidence of IFD was significantly higher in allo-HSCT than it was in auto-HSCT recipients (9.2% versus 3.5%; *p* = 0.001), with a significantly higher mortality risk as well. In our study, 8 out of 10 allo-HSCT patients experienced proven/probable IFD, while 2 out of 6 auto-HSCT patients had possible IFD, which is consistent with the aforementioned studies. Unlike the 6 auto-HCT patients who all achieved a clinical response, only 4 out of 10 allo-HCT patients achieved a clinical response. This suggests that allo-HSCT is not only a risk factor for developing IFD but also for influencing the prognosis of IFD.

Cattaneo et al. [17] conducted a study on the use of ISA among hematological patients. The results showed that there was a better response to ISA among female patients compared to male patients. At the same time, another study [41] showed lower ISA clearance in women, particularly in elderly patients, which may be responsible for the higher plasma levels of ISA in women with subsequent better efficacy. Thus, it seems difficult to reconcile the contrary results in our study with those presented in the aforementioned studies. Faced with such results, we tentatively attempted to attribute the causes to race, population, the distribution of clinical characteristics or disease subtypes between genders, and this study’s small sample size. Further research is needed to expand this drug’s application findings in the Chinese population.

Our study has limitations. First, the sample size was small, and selection bias seemed inevitable. Second, the absence of a control group affected our ability to determine the clinical efficacy and safety of ISA. Again, this observational study relied primarily on historical electronic medical record data, which were not as comprehensive and up-to-date as they should be. Therefore, more prospective studies, as well as studies incorporating pediatric patients, are needed to validate the efficacy and safety of ISA, especially when used in hematology or as a combination therapy.

## 5. Conclusions

To the best of our knowledge, this is the first time that real-world data about the use of ISA in Chinese populations has been described. Using clinical data from previous studies and the preliminary data within the limited scope of this study, we have confirmed the efficacy and safety of ISA in the Chinese population. Furthermore, it is important to note that certain provisional conclusions still need to be further validated through high-level evidence obtained from prospective clinical trials.

## Figures and Tables

**Figure 1 microorganisms-11-02229-f001:**
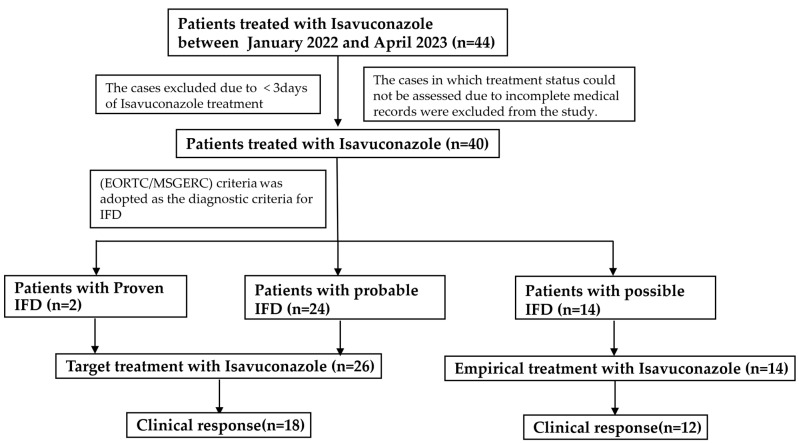
Flowchart of patients included and excluded in this study.

**Figure 2 microorganisms-11-02229-f002:**
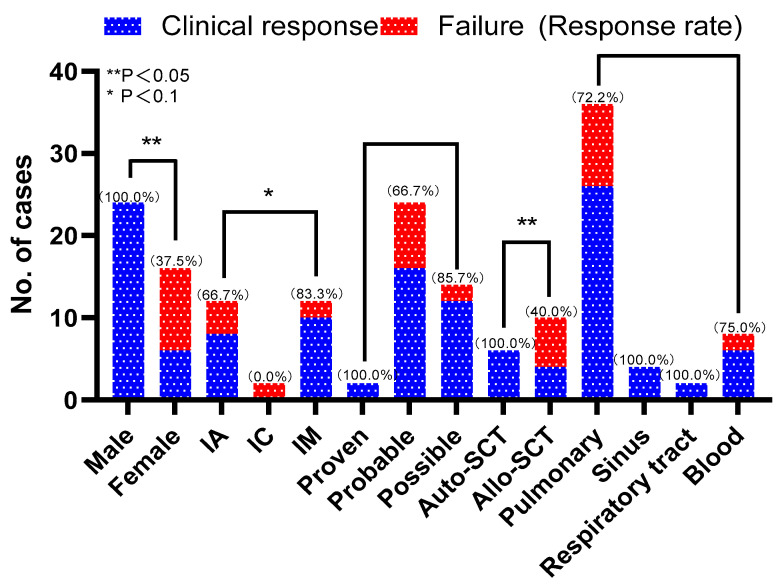
Distribution of clinical responses and failures in different subgroups of cases.

**Table 1 microorganisms-11-02229-t001:** Main demographic and clinical characteristics of the study patients.

Characteristics	Value
**Age, y, median (range) **	45.8 (9–72)
**Male sex, *n* (%)**	16 (40%)
**Weight, Kg, median (range) **	59 (35–96)
**Disease type (%)**	
Diffuse large B-cell lymphoma	8 (20%)
Acute myeloid leukemia	10 (25%)
Natural/killer T-cell lymphoma	2 (5%)
Follicular lymphoma	2 (5%)
Aplastic anemia	2 (5%)
T-cell lymphoblastic lymphoma	2 (5%)
Mixed-phenotype acute leukemia	2 (5%)
Hemophagocytic syndrome	4 (10%)
Sepsis	2 (5%)
Fungal sinusitis	2 (5%)
Pulmonary mycosis	4 (10%)
**IFD**	
Invasive aspergillosis	12 (30%)
Invasive candidiasis	2 (5%)
Invasive mucormycosis	12 (30%)
Unknown	14 (35%)
**Morbidities**	
Diabetes mellitus	10 (25%)
Hepatic insufficiency	8 (20%)
Renal insufficiency	16 (40%)
Graft-versus-host disease	6 (15%)
Respiratory failure	4 (10%)
Autologous	6 (15%)
Allogeneic	10 (25%)
Site of infection	
Pulmonary	36 (90%)
Paranasal sinuses	4 (10%)
Respiratory tract	2 (5%)
Bloodstream infections	6 (15%)
**Classifications—IFD**	
Proven	2 (5%)
Probable	24 (60%)
Possible	14 (35%)
**Outcome**	
**Diagnosis of IFD**	
Partial remission	30 (75%)
Progression	7 (10%)
Dead	3 (15%)
**Type of IFD**	
Invasive aspergillosis	8/12 (66.7%)
Invasive candidiasis	0/2 (0%)
Invasive mucormycosis	10/12 83.3%)
**Site of infection**	
Pulmonary	26/36 (72.2%)
Paranasal sinuses	4/4 (100%)
Respiratory tract	2/2 (100%)
Bloodstream infections	6/8 (75%)
**Therapy**	
Target treatment	18/26 (69.2%)
Empirical treatment	12/14 (85.7%)

**Table 2 microorganisms-11-02229-t002:** Characteristics of IFD cases.

NO	Gender	Age	Disease Type	Outcome	Site	Key Evidence	Diagnosis
1	Female	59	AML	dead	pulmonary	CT; *Candida albicans* determined via phlegm culture	Probable
2	Male	43	hemophagocytic syndrome	partial remission	pulmonary	CT; febrile neutropenia	Possible
3	Female	42	AML	progression	pulmonary	*Aspergillus* determined via BAL fluid culture	Probable
4	Male	54	natural/killer T-cell lymphoma	partial remission	pulmonary	*Aspergillus* determined via blood mNGS; CT	Probable
5	Male	72	DLBCL	partial remission	pulmonary/respiratory tract	CT; *Aspergillus* determined via phlegm culture	Probable
6	Female	43	aplastic anemia	dead	pulmonary/bloodstream infections	*Rhizomucor pusillus* determined via blood mNGS; CT	Probable
7	Male	52	DLBCL	partial remission	pulmonary	CT; febrile neutropenia	Possible
8	Male	22	AML	partial remission	pulmonary	CT; febrile neutropenia	Possible
9	Female	70	DLBCL	partial remission	pulmonary	CT; febrile neutropenia	Possible
10	Female	48	AML	progression	pulmonary	CT; *Aspergillus* determined via phlegm culture	Probable
11	Male	38	T-cell lymphoblastic lymphoma	partial remission	pulmonary	CT; *Aspergillus* determined via BAL fluid mNGS	Probable
12	Male	57	sepsis	partial remission	pulmonary/bloodstream infections	*Rhizopus oryzae* determined via mNGS; CT	Probable
13	Male	20	mixed phenotype acute leukemia	partial remission	pulmonary	*Aspergillus* determined via BAL fluid culture; CT	Probable
14	Male	9	AML	partial remission	pulmonary	CT; febrile neutropenia	Possible
15	Female	40	hemophagocytic syndrome	partial remission	pulmonary/bloodstream infections	*Rhizomucor* determined via mNGS; CT	Probable
16	Female	66	DLBCL	dead	pulmonary	CT; fever	Possible
17	Male	61	follicular lymphoma	partial remission	Bloodstream infections/paranasal sinuses	CT; *Lichtheimia corymbifera* determined via mNGS	Probable
18	Female	33	pulmonary mycosis	partial remission	pulmonary	History of *Mycosis* infection	Possible
19	Male	52	fungal sinusitis	partial remission	paranasal sinuses	Histopathology showing true hyphae (IM)	Proven
20	Male	48	pulmonary mycosis	partial remission	pulmonary	CT; *Rhizopus delemar* identified using fluid mNGS	Probable
21	Female	58	DLBCL	progression	pulmonary	CT; *Candida albicans* determined via phlegm culture	Probable
22	Male	44	AML	partial remission	pulmonary	CT; febrile neutropenia	Possible
23	Male	50	hemophagocytic syndrome	partial remission	pulmonary	CT; Fever	Possible
24	Female	40	AML	progression	pulmonary	*Aspergillus* determined via BAL fluid culture	Probable
25	Male	56	natural/killer T-cell lymphoma	partial remission	pulmonary	Blood GM test positive 1 time (IA); CT	Probable
26	Male	16	AML	partial remission	pulmonary	CT; febrile neutropenia	Possible
27	Male	72	hemophagocytic syndrome	partial remission	pulmonary/respiratory tract	*Rhizopus oryzae* determined via mNGS; CT	Probable
28	Female	45	aplastic anemia	progression	pulmonary/bloodstream infections	*Rhizomucor* determined via mNGS; CT	Probable
29	Male	48	pulmonary mycosis	partial remission	pulmonary	CT; *Rhizopus delemar* identified using fluid mNGS	Probable
30	Male	55	sepsis	partial remission	pulmonary/bloodstream infections	CT; *Rhizomucor* determined via mNGS	Probable
31	Male	22	mixed phenotype acute leukemia	partial remission	pulmonary	CT; febrile neutropenia	Possible
32	Male	24	AML	partial remission	pulmonary	CT; febrile neutropenia	Possible
33	Female	68	AML	partial remission	pulmonary	*Aspergillus* determined via BAL fluid culture; CT	Probable
34	Female	50	fungal sinusitis	progression	pulmonary	CT; Fever	Possible
35	Male	40	T-cell lymphoblastic lymphoma	partial remission	pulmonary	Histopathology showing true hyphae (IM)	Proven
36	Male	30	DLBCL	partial remission	paranasal sinuses	CT; *Aspergillus* determined via BAL fluid mNGS	Probable
37	Female	44	DLBCL	partial remission	pulmonary/Bloodstream infections	CT; *Aspergillus* determined via phlegm culture	Probable
38	Female	53	follicular lymphoma	progression	pulmonary	CT; *Aspergillus* determined via phlegm culture	Probable
39	Male	36	DLBCL	partial remission	Bloodstream infections/paranasal sinuses	CT; *Rhizomucor* determined via mNGS	Probable
40	Female	55	pulmonary mycosis	partial remission	pulmonary	History of *Mycosis* infection	Possible

AML: Acute myeloid leukemia. DLBCL: Diffuse large B-cell lymphoma.

**Table 3 microorganisms-11-02229-t003:** Characteristics of patients presenting clinical response or clinical failure.

Characteristics	Clinical Response (*n*/Median, IQR)	Clinical Failure (*n*/Median, IQR)	*p*
Age	44.4 (9–72)	50.4 (40–66)	0.235
Gender (male/female)	24/6	0/10	0.000 **
Allogeneic/autologous	6/4	0/6	0.027 **
Target treatment/empirical treatment	18/12	8/2	0.446

(** *p* < 0.05).

## Data Availability

The data that support the findings of this study are available from the corresponding author upon reasonable request. The data are not publicly available due to privacy and ethical constraints.

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
