# Peer review of "Real-World, Multicenter Case Series of Patients Treated with Isavuconazole for Invasive Fungal Disease in China"

_microorganisms, 2023, doi:10.3390/microorganisms11092229_

Round 1
Reviewer 1 Report
In this article, authors attempt to describe the effectiveness of isavuconazole in the real world for patients suffering from invasive fungal disease in China.
In general, the paper is well written, and English is good. There are some exceptions to that, and I am underlining them below.
The title describes well the purpose of the study and the same goes for the abstract.
Introduction describes in a satisfactory way the data about the way of action of isavuconazole and the up-to-date data about its use.
Materials and methods are clearly presented and results too.
RESULTS COMMENTS:
Lines 126 - 127 . Please rephrase paying attention to the verbs.
Table 2 . patient number 10. It is extremely rare to culture Aspergillus from patients ‘blood. Was this information correct?
Phlegm culture . Perhaps another way to put it would be preferable.
Line 140. Clinical response patients . please rephrase
Line 141. What do you mean in male was higher than in females? Please rephrase
Line 143. Do not start the sentence with :as well as” rephrase so that the meaning is more clear.
Line 144. At the same time///separately. Note clear meaning , please rephrase.
Line 153. Patients cannot “fail to obtain etiological evidence”.
Line 160. Please rephrase, something is missing.
Line 176. Had tended to increase : wrong English tense
Line 209. Trial trial
Line 213. Predictably…….there is something missing in this sentence
Line 261. Which information logging ..needs better English
CONCLUSIONS : has been statistically described. What does this mean? Express it in another way.
“ And the observational validation “ please express this in another way.
You have reason to believe that …….
Why specially in Chinese people? Is there a reason why Chinese people would be different ?
Line 269. To be further confirmed.
English is generally good but need amelioration
Author Response
Responses to Reviewer 1:
- Question:
Lines 126 - 127. Please rephrase paying attention to the verbs.
Line 140. Clinical response patients. please rephrase.
Line 141. What do you mean in male was higher than in females? Please rephrase.
Line 143. Do not start the sentence with: as well as” rephrase so that the meaning
is clearer.
Line 144. At the same time///separately. Note clear meaning, please rephrase.
Line 153. Patients cannot “fail to obtain etiological evidence”.
Line 160. Please rephrase, something is missing.
Line 176. Had tended to increase: wrong English tense.
Line 209. Trial trial.
Line 213. Predictably…….there is something missing in this sentence.
Line 261. Which information logging. needs better English.
CONCLUSIONS: has been statistically described. What does this mean? Express it in another way.
“And the observational validation” please express this in another way.
Response: Thank you for your meticulous suggestions. We have revised the above errors in the revised manuscript. In addition, the language of the manuscript has been reviewed and polished throughout. The revised manuscript with the track changes has been submitted.
- Question: Table 2. patient number 10. It is extremely rare to culture Aspergillus from patients ‘blood. Was this information correct? Phlegm culture. Perhaps another way to put it would be preferable.
Response: Thank you for your professional comments. We are so sorry for that mistake; the aspergillus was cultured from phlegm and we have corrected this error in the revised manuscript. Thanks once again for your suggestions.
- Question: You have reason to believe that …….Why specially in Chinese people? Is there a reason why Chinese people would be different? Line 269. To be further confirmed.
Response: Thank you for your professional comments. Given that ISA (isavuconazole) was only approved in China at the end of 2021 for the treatment of IA and IM in adults, there is a lack of early clinical use experience and data. Ethnic information does have a certain influence on drug efficacy. Indeed, in the subsequent efficacy studies, we also found different results from the previous European and American populations, such as the impact of gender on prognosis.

Reviewer 2 Report
This manuscript describes a real-world, multicenter study conducted in China on the use of isavuconazole (ISA) for the treatment of invasive fungal diseases in immunocompromised patients. The topic is interesting but it is still suboptimal.
Aspects that need to be improved in the manuscript:
1. Introduction:
- The introduction lacks a clear statement of the research problem and objectives.
- It would be beneficial to provide more context and background information on invasive fungal diseases (IFD) and the significance of the research. (Invasive fungal diseaseS, in plural, there is not not only one, but several, and this is a major aspect that has been comprehensively ingnored.
- The introduction can be structured better with a clear transition from general information about IFD to specific information about the study's focus on ISA (isavuconazole) as a new treatment option.
2. Materials and Methods:
- The methods section could be more detailed, providing information about the study design, inclusion and exclusion criteria, and patient selection process in a clearer manner.
- The authors should specify the reason for selecting the three medical centers in Hubei, China, and discuss the implications of this selection on the study's generalizability. Also mention which hospitals are there and their characteristics.
- The diagnostic criteria for IFD (invasive fungal disease) need to be clearly stated with references to the specific criteria used.
3. Results:
- The presentation of the results could be improved by using tables and figures to display the data more effectively.
- The characteristics of the patients (age, gender, disease type, etc.) can be presented in a clear and concise manner using a table.
- The clinical response rates and outcomes for different subgroups of patients (e.g., by infection type, proven/probable/possible IFD) should be presented in a more organized format.
- The interpretation of the results should be strengthened by discussing the clinical significance of the findings and comparing them with existing literature.
- Authors need to keep in mind that there is not only 1 IFD, but many. Besides, ISA is not the first line for Candida spp. (echonicandins) or Mucorales (l-AMB is preferred), so please, do not use such strong statements
4. Discussion:
- The discussion section needs to be expanded to include a more comprehensive review of the literature and a comparison of the study's findings with previous research.
- The limitations of the study should be clearly acknowledged, including the small sample size, potential selection bias, and lack of a control group.
- Future research directions and potential areas of investigation should be mentioned to provide readers with insights into furthering the understanding of ISA's efficacy and safety.
5. General:
- The manuscript should be carefully proofread to eliminate grammatical errors and improve the overall writing clarity.
- Citations and references should be properly formatted following the journal's guidelines.
- Considering this manuscript, it seems that the authors categorized all fungal infections together without distinguishing between different types of fungal pathogens or their specific clinical implications. Treating all fungal infections as the same may oversimplify the complexity and diversity of these infections, as different fungi can have varying pathogenicity, treatment responses, and clinical outcomes. It would be more informative and clinically relevant to analyze and report the data separately for specific fungal pathogens to provide a more nuanced understanding of the effectiveness of isavuconazole in treating each type of infection.
Author Response
Responses to Reviewer 2:
- Question: Introduction:
The introduction lacks a clear statement of the research problem and
objectives. It would be beneficial to provide more context and background information on invasive fungal diseases (IFD) and the significance of the research. (Invasive fungal diseases, in plural, there is not not only one, but several, and this is a major aspect that has been comprehensively ingnored.
The introduction can be structured better with a clear transition from general information about IFD to specific information about the study's focus on ISA (isavuconazole) as a new treatment option.
Response: Thank you for your professional comments. More statement of the background and purpose of the research as well as refined information of the categories of IFD were described in the abstract and the introduction. Thanks once again for your suggestions.
Question: Materials and Methods:
- The methods section could be more detailed, providing information about the study design, inclusion and exclusion criteria, and patient selection process in a clearer manner.
- The authors should specify the reason for selecting the three medical centers in Hubei, China, and discuss the implications of this selection on the study's generalizability. Also mention which hospitals are there and their characteristics.
- The diagnostic criteria for IFD (invasive fungal disease) need to be clearly stated with references to the specific criteria used.
Response: Thank you for your professional comments. We have added a flowchart into the manuscript which could be more detailed, providing information about the study design, inclusion and exclusion criteria, and patient selection process in a clearer manner, See the fig.1 for details.
The selected three medical centers are all general teaching hospitals in Hubei Province. They are the first three hospitals in Hubei province using ISA. The revised 2020 European Organization for Research and Treatment of Cancer and the Mycoses Study Group Education and Research Consortium (EORTC/MSGERC) criteria [see reference 21for details] was adopted as the diagnostic criteria for IFD. Thanks once again for your suggestions.
- Question: Results:
- The presentation of the results could be improved by using tables and figures to display the data more effectively.
- The characteristics of the patients (age, gender, disease type, etc.) can be presented in a clear and concise manner using a table.
- The clinical response rates and outcomes for different subgroups of patients (e.g., by infection type, proven/probable/possible IFD) should be presented in a more organized format.
- The interpretation of the results should be strengthened by discussing the clinical significance of the findings and comparing them with existing literature.
- Authors need to keep in mind that there is not only 1 IFD, but many. Besides, ISA is not the first line for Candida spp. (echonicandins) or Mucorales (l-AMB is preferred), so please, do not use such strong statements.
Response: Thank you for your professional comments.
The additional information has been added into as detailed in the table 1.
The final results of this study and the clinical significance of comparing them with the existing literature to enhance the interpretation of the results have been discussed in the discussion section of the revised manuscript. At the same time, some of the strong statement descriptions have been modified.
- Question: Discussion:
- The discussion section needs to be expanded to include a more comprehensive review of the literature and a comparison of the study's findings with previous research.
- The limitations of the study should be clearly acknowledged, including the small sample size, potential selection bias, and lack of a control group.
- Future research directions and potential areas of investigation should be mentioned to provide readers with insights into furthering the understanding of ISA's efficacy and safety.
Response: Thank you for your professional comments. All the results of this study have been discussed in comparison with previously available studies in the revised manuscript. The discussion was modified with explanation and acknowledgement of the limitations of the study. Thanks once again for your suggestions.
- Question: General:
- The manuscript should be carefully proofread to eliminate grammatical errors and improve the overall writing clarity.
- Citations and references should be properly formatted following the journal's guidelines.
- Considering this manuscript, it seems that the authors categorized all fungal infections together without distinguishing between different types of fungal pathogens or their specific clinical implications. Treating all fungal infections as the same may oversimplify the complexity and diversity of these infections, as different fungi can have varying pathogenicity, treatment responses, and clinical outcomes. It would be more informative and clinically relevant to analyze and report the data separately for specific fungal pathogens to provide a more nuanced understanding of the effectiveness of isavuconazole in treating each type of infection.
Response: Thank you for your professional comments. The language of the manuscript has been reviewed and polished throughout; Citations and references have been properly formatted following the journal's guidelines. We have added the data separately about Type of IFD and Site of infection into the table 1. The purpose of this study is to describe the early experience and efficacy of ISA in Chinese population, in order to provide references for clinical rational use of ISA. The sample size was not enough to statistically describe the clinical efficacy of detail different sites in different type of IFD infections. Therefore, this was included as a limitation of our study in the revised manuscript. Thanks once again for your suggestions.
